# Identification of Immunogenic Cell Death-Related Signature for Glioma to Predict Survival and Response to Immunotherapy

**DOI:** 10.3390/cancers14225665

**Published:** 2022-11-18

**Authors:** Zhiqiang Sun, Hongxiang Jiang, Tengfeng Yan, Gang Deng, Qianxue Chen

**Affiliations:** 1Department of Neurosurgery, Renmin Hospital of Wuhan University, Wuhan 430060, China; sunzq12138@whu.edu.cn (Z.S.); hx-jiang@whu.edu.cn (H.J.); 2Central Laboratory, Renmin Hospital of Wuhan University, Wuhan 430060, China; 3Department of Neurosurgery, The Second Affiliated Hospital of Nanchang University, Nanchang 330006, China; 2019103020026@whu.edu.cn; 4Institute of Neuroscience, Nanchang University, Nanchang 330031, China

**Keywords:** glioma, immunogenic cell death, overall survival, immunotherapy

## Abstract

**Simple Summary:**

Glioma is a malignant primary brain tumor accounting for 75% of the total cases. Notably, several immunotherapeutic and chemotherapeutic agents in glioma have been demonstrated to induce immunogenic cell death (ICD). However, the studies on glioma have targeted individual ICD-related genes, and comprehensive analyses of all the ICD-related genes are still lacking. The aim of this study was to identify a novel molecular signature based on ICD-related genes in glioma, which might be beneficial for the diagnosis and treatment of glioma. We eventually identified a 14 ICD-related gene signature, which could effectively predict prognosis and immunotherapy response in glioma. These findings might be significantly helpful in selecting the best therapeutic strategy for glioma.

**Abstract:**

Immunogenic cell death (ICD) is a type of regulated cell death (RCD) and is correlated with the progression, prognosis, and therapy of tumors, including glioma. Numerous studies have shown that the immunotherapeutic and chemotherapeutic agents of glioma might induce ICD. However, studies on the comprehensive analysis of the role of ICD-related genes and their correlations with overall survival (OS) in glioma are lacking. The genetic, transcriptional, and clinical data of 1896 glioma samples were acquired from five distinct databases and analyzed in terms of genes and transcription levels. The method of consensus unsupervised clustering divided the patients into two disparate molecular clusters: A and B. All of the patients were randomly divided into training and testing groups. Employing the training group data, 14 ICD-related genes were filtered out to develop a risk-score model. The correlations between our risk groups and prognosis, cells in the tumor microenvironment (TME) and immune cells infiltration, chemosensitivity and cancer stem cell (CSC) index were assessed. A highly precise nomogram model was constructed to enhance and optimize the clinical application of the risk score. The results demonstrated that the risk score could independently predict the OS rate and the immunotherapeutic response of glioma patients. This study analyzed the ICD-related genes in glioma and evaluated their role in the OS, clinicopathological characteristics, TME and immune cell infiltration of glioma. Our results may help in assessing the OS of glioma and developing better immunotherapeutic strategies.

## 1. Introduction

Immunogenic cell death (ICD) is a type of regulated cell death (RCD) and an important mechanism of antitumor immunity [1,2]. Numerous studies have investigated the underlying mechanism of ICD and showed that the immunogenic features of ICD were majorly mediated by damage-associated molecular patterns (DAMPs), incorporating surface-exposed calreticulin (CRT), released high mobility group 1 (HMGB1) protein, and secreted adenosine triphosphate (ATP) [3,4,5,6]. Cancer immunotherapy activates the immune cells to kill tumor cells. Numerous studies have demonstrated that ICD can activate tumor immunity [7]. In addition, numerous preclinical models have applied ICD for tumors [8,9,10,11]. However, studies on the applications of ICD in clinics are limited. Therefore, the applications of ICD in clinics should be further investigated. In particular, the novel biomarkers of ICD should be explored to aid in the individual therapy of patients.

Glioma is a malignant primary brain tumor, accounting for 75% of the total cases; there are about 10,000 patients diagnosed annually [12,13,14]. The World Health Organization (WHO) 2021 CNS tumors classification recommended the diagnosis of CNC tumors based on molecular markers, indicating the importance of glioma molecular therapy [15]. Notably, several immunotherapeutic and chemotherapeutic agents in glioma have been demonstrated to induce ICD [8,16], which is an important factor affecting the therapy responses in glioma [17]. Additionally, although there have been some studies on ICD-related genes in glioma, these studies only focused on lower-grade gliomas (LGG) or glioblastoma (GBM), and comprehensive and systematic studies on ICD-related genes in grade II to IV gliomas are still lacking [18,19]. Therefore, novel molecular markers based on ICD-related genes in grade II to IV gliomas, which might be beneficial for the diagnosis and treatment of glioma, should be further explored.

In this study, ICD-related molecular clusters and a risk model of ICD, which could effectively predict prognosis and immunotherapy response in glioma, were constructed. These findings may be significantly helpful in selecting the best therapeutic strategy for glioma.

## 2. Materials and Methods

### 2.1. Datasets

The study scheme is presented in Appendix A. The transcriptomic profiles (transcripts per kilobase million, TPM) and related clinical data of glioma were acquired from The Cancer Genome Atlas (TCGA, https://portal.gdc.cancer.gov/, (accessed on 4 May 2022)) and China Glioma Genome Atlas (CGGA, http://www.cgga.org.cn/, (accessed on 5 May 2022)) databases. Two mRNA-seq datasets (mRNAseq_325 and mRNAseq_693) and one microarray dataset of CGGA glioma cohorts were downloaded from CGGA, while the TCGA glioma cohorts included TCGA-lower-grade glioma (LGG) and TCGA-glioblastoma (GBM) datasets. Subsequently, the batch effects of these five datasets were normalized using the “sva” package. The transcriptomic profiles and clinicopathological data of the five datasets were then combined in R (version 4.1.2, https://www.r-project.org, (accessed on 12 November 2021)). Finally, a total of 1896 glioma patients were selected in this study for subsequent analysis. Their details are listed in Appendix A.

### 2.2. Consensus Clustering

We selected 34 ICD-related genes from the literature [20]. After intersecting these 34 genes with the transcriptomic profiles, 31 ICD-related genes remained. The details of ICD-related genes are presented in Appendix A. A total of 1896 patients were classified into two different clusters using the R package “ConsensusClusterPlus”, using the expression values of ICD-related genes. The best clustering was identified based on the following standards: a smooth increase in the cumulative distribution function (CDF) curve, a sufficiently large size of all the groups, an increase in the correlations within the group, and a decrease in the correlations between the groups. Next, we acquired a hallmark gene set (c2.cp.kegg.v7.5.1.symbols.gmt) from the gene set enrichment analysis (GSEA, http://www.gsea-msigdb.org/gsea/index.jsp, (accessed on 18 May 2022)) to perform a gene set variation analysis (GSVA). A heatmap of the top 20 genes was plotted using the R package “pheatmap”.

### 2.3. Glioma in Different Molecular Clusters: Clinical Features and Prognosis

After clustering, the two molecular clusters were evaluated based on the clinical features to investigate the correlations between prognosis, molecular clusters, and clinicopathological features. The clinicopathological features included age, gender, grade, histology, PRS type, TCGA type, *IDH* mutation status, 1p19q codeletion status, MGMT methylation status, and chemoradiotherapy status. Moreover, the differences between the two clusters were assessed using the Kaplan–Meier (K–M) analysis, using “survival” and “survminer” in R.

### 2.4. Correlations of Molecular Clusters with TME and Immune Checkpoint Genes

The TME of every glioma patient was evaluated using the R package “estimate”. Furthermore, using the “CIBERSORT” algorithm [21], the levels of 22 human immune cell subsets in each glioma patient was calculated. The correlations between 30 immune checkpoint genes, including PDCD1 (also known as PD-1) and CD274 (also known as PD-L1), and the two clusters, were analyzed respectively.

### 2.5. Differentially Expressed Genes (DEGs) Identification and Functional Enrichment Analysis

The DEGs between the two clusters were filtered employing “limma” with the cut-off values of *p*-value < 0.05 and absolute log2 fold-change (log2 FC) > 0.585. Then, we employed the package “clusterprofiler” to conduct a functional genes and pathways enrichment analysis.

### 2.6. Development of an ICD-Related Risk-Score Model

The ICD pattern of every sample, based on their risk score, was presented. First, the correlations between DEGs and glioma survival were determined using univariate Cox regression analyses of DEGs. Second, using an unsupervised clustering technique, the DEGs were divided into two groups (DEGs cluster A and DEGs cluster B) based on the ICD-related genes, which were associated with glioma prognosis. In the last step, the ICD-related risk scores were calculated, and we randomly grouped the patients into two groups: the training group (*n* = 916) and the testing group (*n* = 916). Using the training group data, we developed the risk-score model. In general, the overfitting risk in the LASSO Cox regression model was minimized using the “glmnet”, and the prognostic genes were filtered in the training group using multivariate Cox analyses. The risk scores were calculated using Equation (1).
Risk-score = Σ (expression × coef)(1)
where “expression” denoted the expression value of genes, and the “coef” denoted the risk coefficient of genes. The patients with risk scores lower than the median risk score were divided into the low-risk group, while those with risk scores higher than the median were divided into the high-risk group. Then, we conducted a K–M analysis to assess the differences between our two groups. Moreover, the differences between the distribution of our two groups were analyzed using principal component analysis (PCA). Moreover, the testing group and other external datasets were similarly analyzed.

### 2.7. Tissue Samples

Six tumor samples were acquired from glioma patients, and six non-tumor samples were acquired from patients with cerebral hemorrhage. All of the samples were stored at −80 °C. Our study was approved by the Ethics Committee of the Renmin Hospital of Wuhan University. The patients were informed about the purpose of this study and signed an informed consent form.

### 2.8. RNA Extracting and Quantitative Real-Time PCR

The extraction of total RNA from the tissues was conducted using TRIzol reagent (Invitrogen, Carlsbad, CA, USA). The extracted RNA was reverse transcribed into cDNA using the PrimeScript RT Reagent Kit (RR047A, Takara, Japan). The mRNA expression levels of our signature genes in the two groups were analyzed with SYBR-Green assays (Takara) using the CFX-96 instrument (Bio-Rad Laboratories, Inc., Hercules, CA, USA). The relative mRNA expression levels of the genes were calculated using the 2^−∆∆Ct^ method, and the *GAPDH* gene was selected as the internal control. Our primer sequences are listed in Appendix A.

### 2.9. Clinical Relevance and Stratified Analysis of the Risk-Score

The correlations between the risk score and clinicopathological characteristics (age, gender, grade, histology, PRS type, TCGA type, *IDH* mutation, 1p19q codeletion, MGMT methylation, and chemoradiotherapy) were analyzed using the chi-squared test. Univariate and multivariate regression analyses were used to assess the potential of using risk-score as an independent prognostic factor of the patients in the training and testing groups. Additionally, based on the various clinical features, we grouped patients into different subgroups. The predictive potential risk scores were evaluated in each subgroup using the K–M curve analysis.

### 2.10. Immune Status and Cancer Stem Cell (CSC) Index in Different Risk Groups

The number of the 22 infiltrated immune cell types in each case was calculated using CIBERSORT. The correlations of these 22 immune cell types with 14 risk-score model-related genes were assessed in a heatmap. The differences in immune checkpoint gene expression levels between the low- and high-risk groups, and between the ICD-related gene clusters, were determined using the R package “ggplot2”. Moreover, a scatter plot was drawn to determine the correlations between the risk score and CSC.

### 2.11. Drug Sensitivity and Mutation Analysis

The somatic mutations in the TCGA-GBM and TCGA-LGG datasets were acquired from the TCGA database. Then, we measured the tumor mutation burden (TMB) of each patient and assessed the difference between the low- and high-risk groups. We also evaluated the correlations between our risk score and TMB. Waterfall plots were drawn using the “maftools” package to identify the top-20 mutated genes in glioma between the two risk groups. Finally, we calculated the half-inhibitory concentration (IC50) values of the glioma chemotherapeutic agents using the “pRRophetic” package to assess the chemotherapeutic response in the two risk groups.

### 2.12. Development of the Nomogram

A nomogram model was developed using the “rms” package. The nomogram model was characterized by WHO grades and the risk scores, to enhance the clinical applications of the risk score model. We assigned a score to each sample and summed the scores of each sample to get the nomogram score [22]. Then, we assessed the predictive accuracy of our nomogram model by plotting the 1-, 3-, 5-, and 10-year receiver operating characteristic (ROC) curves. The differences between the predicted 1-, 3-, 5-, and 10-year survival rates were determined using the calibration plots.

### 2.13. Statistical Analyses

All the data were analyzed using the R package. Cox regression and K–M survival analyses were performed to examine the correlations between our groups and overall survival (OS) time. The comparisons of the two groups were conducted using the Wilcox test. Kruskal-Wallis test was chosen to analyze the correlations of clinicopathological characteristics with the groups. A *p*-value of <0.05 was considered statistically significant.

## 3. Results

### 3.1. Transcriptional Changes of ICD-Related Genes in Glioma

A total of 31 ICD-related genes were analyzed in this study. As shown in Figure 1A,B, the 31 ICD-related genes showed a higher overall somatic mutation rate in the GBM dataset as compared to those in the LGG dataset. Figure 1A shows the waterfall plot of mutation frequency in the ICD-related genes in the 523 LGG samples, containing 65 (12.43%) mutated samples. In the ICD-related gene set, most of the genes were unmutated. The *PIK3CA* gene had the highest mutation frequency (9%), followed by the *ATG5* (1%), *CASP8* (1%), and *ENTPD1* (1%) genes. Moreover, the GBM group showed a higher mutation rate (18.66%, 86/461 samples) compared with the LGG group (Figure 1B). However, the gene with the highest mutation frequency was still *PIK3CA*, followed by *NLRP3* (3%), *CASP1* (1%), *FOXP3* (1%), *IFNG* (1%), *CD8A* (1%), and *LY96* (1%).

Then, the changes in the copy number in the ICD-related genes were also detected between the LGG and GBM groups; most of them showed copy number variations (CNVs). Among the 31 ICD-related genes, the copy numbers of *IFNA1*, *IFNB1*, *BAX*, *IFNGR1*, *HMGB1*, *HSP90AA1*, and *PDIA3* decreased, while those of *CD4*, *PIK3CA*, and *P2RX7* increased (Figure 1C). The positions of ICD-related genes on the 23 pairs of chromosomes are presented in Figure 1D. The changes in the expression levels of most ICD-related genes between glioma and the normal brain tissues were positively correlated with the copy numbers. As compared to the normal brain tissues, the genes with decreased copy numbers, such as *IFNA1*, *IFNB1*, and *IFNG*, had lower expression levels in glioma. Similarly, the genes with increased copy numbers, such as *CD4*, *PIK3CA*, and *P2RX7*, had higher expression levels in glioma (Figure 1E), indicating a positive correlation of CNVs with the expression levels of ICD-related genes. Nevertheless, some genes, such as *BAX*, *IFNGR1*, and *PDIA3*, showed a decrease in their copy number and exhibited elevated expression levels in glioma. This indicated that the regulatory mechanisms of these genes’ expression were complicated and diverse. CNV was one of these regulatory mechanisms, while transcription factors, DNA methylation, etc. might also be involved in the regulation of these genes [23,24]. These results suggested a notable change at both transcriptional and genetic levels of ICD-related genes in glioma as compared to the normal brain tissues, indicating that ICD-related genes might be correlated with the progression of glioma.

### 3.2. Construction of ICD-Related Gene Clusters in Glioma

Five different glioma datasets, including CGGA-mRNAseq_325, mRNAseq_693, microarray, TCGA-LGG, and TCGA-GBM datasets, were integrated. After filtration, a total of 1896 patients were left. Their details are listed in Appendix A. In order to evaluate the prognostic potential of the 31 ICD-related genes in glioma, the K–M and univariate Cox analyses were performed with filter criterion *p* < 0.05, resulting in 24 prognostic genes (Appendix A). Then, a multivariate Cox analysis was immediately conducted to filter out the independent prognostic genes, resulting in 12 ICD-related genes, including *BAX*, *CALR*, *CASP1*, *CASP8*, *CD4*, *IL10*, *IL1R1*, *IL6*, *LY96*, *MYD88*, *NT5E*, and *PDIA3* (Table 1). In the gene network, we comprehensively demonstrated the correlations and interactions of the ICD-related genes and their impacts on the prognosis of glioma patients (Figure 2A and Appendix A). Furthermore, a consensus clustering method was used to group the glioma patients using the integrated transcriptional data of 31 ICD-related genes to deeply investigate their expression features in glioma (Appendix A). In Appendix A, the consensus matrix heatmaps are presented, showing k = 2 as the best choice. Based on this criterion, the patients were categorized into cluster A (*n* = 630) and cluster B (*n* = 1266) (Figure 2B). The prognostic potential in these clusters was evaluated using PCA analysis, which illustrated a significant clustering effect between the two groups (Figure 2C). As shown in Figure 2D, the K–M curve presented a better prognostic potential in cluster B as compared with that in cluster A (*p* < 0.001). The heatmap revealed the distribution of clinical characteristics in different clusters, which suggested that, as compared to cluster B, cluster A was related to the higher WHO-grade glioma without *IDH* mutations. Among the two clusters, the expression levels of ICD-related genes indicated that most of the ICD-related genes were upregulated and downregulated in clusters A and B, respectively (Figure 2E).

### 3.3. TME Characteristics Differ between Clusters

GSVA analyses were conducted to investigate the functional pathways in two clusters. The results suggested that the immune-related pathways, including natural killer cell-mediated cytotoxicity, antigen processing and presentation, cytokine receptor interaction, leukocyte trans-endothelial migration, Toll-like receptor, and NOD-like receptor signaling pathways, were closely related to cluster A (Figure 3A; Appendix A). The infiltration of 22 immune cell types in every glioma sample (Appendix A) was calculated using a CIBERSORT analysis. Among these cells, types of immune cells were highly distinct in the two clusters (Figure 3B). As compared to cluster B, cluster A exhibited a higher number of M0, M1, and M2 macrophages, neutrophils, CD8+ T cells, and CD4 memory-activated T cells, and a lower number of naive CD4+ T cells, resting CD4 memory T cells, memory and naive B cells, plasma cells, regulatory T cells (Tregs), gamma delta (γδ) T cells, activated NK cells, monocytes, activated dendritic cells, and resting and activated mast cells. Moreover, as shown in Figure 3C,D, the differences in the expression levels of *PDCD1* and *CD274*, which were the two most important immune checkpoint genes in the two clusters, were investigated. The expression levels of other 28 immune checkpoint genes were also evaluated (Appendix A). The TME scores, including the stromal, immune, and ESTIMATE scores of each sample, were calculated using the “estimate” package. The stromal score represented the number of stromal cells, the immune score represented the number of immune cells, and the ESTIMATE score represented the total number of both cell types. As shown in Figure 3E, cluster A exhibited higher scores among all the three scores.

### 3.4. Identifying Gene Clusters

We conducted the gene function enrichment analyses and identified a total of 3551 ICD-related DEGs. The biological function of ICD patterns was further analyzed (Figure 4A,B; Appendix A). The gene ontology (GO) enrichment analysis found that ICD-related DEGs were significantly related to immunological functions (Figure 4A). Meanwhile, the Kyoto Encyclopedia of Genes and Genomes (KEGG) enrichment analyses indicated that these DEGs were related to immune- and tumor-related pathways (Figure 4B), indicating that these genes might be a potential regulating factor in TME. Then, these results were further verified. As presented in Appendix A, the univariate Cox analysis presented a total of 3517 genes that were related to the prognosis of glioma. The samples were then divided into two DEG clusters (A and B) by analyzing 3517 genes using the consensus clustering method (Appendix A). The difference in survival rates in the two clusters was analyzed using the K–M curve analysis, which showed that the survival time of DEG cluster B was longer than that of cluster A (*p* < 0.001; Figure 4C). Furthermore, the results also showed that DEG cluster B was significantly correlated with *IDH* mutations, 1p19q codeletion, and lower WHO grade, while DEG cluster A was correlated with *IDH* wild type, 1p19q non-codeletion, and higher WHO grade (Figure 4D). The differences in the expression levels of ICD-related genes in the two DEG clusters were further analyzed. The results suggested that most of the genes were upregulated in DEG cluster A; most of these genes were downregulated in DEG cluster B (Figure 4E).

### 3.5. Construction of the Prognostic Risk Score

Based on the DEG clusters, the risk-score system was further developed. The patients’ distribution in these groups was presented in a Sankey diagram (Figure 5A). First, using the R package “caret”, we divided patients into the training group (*n* = 916) and the testing group (*n* = 916). Then, using LASSO regression analyses, the 3517 ICD-related DEGs were filtered, showing that a total of 37 genes were filtered out (Appendix A). Among these 37 OS-related genes, the independent prognostic genes were identified using a multivariate Cox analysis. Finally, the optimal signature genes were identified, including 14 genes (*BMP2*, *SEMA3G*, *GCAT*, *SFRP2*, *WNT7B*, *EN1*, *HK2*, *TPX2*, *SMS*, *TFPI*, *CRNDE*, *PLOD3*, *KLF10*, and *ACSL1*), among which, 9 genes (*EN1*, *HK2*, *TPX2*, *SMS*, *TFPI*, *CRNDE*, *PLOD3*, *KLF10*, and *ACSL1*) were high-risk genes and 5 genes (*BMP2*, *SEMA3G*, *GCAT*, *SFRP2*, and *WNT7B*) were low-risk genes (Appendix A). Then, the risk scores were calculated according to Equation (2).
Risk-score = (−0.1569 × BMP2) + (−0.1181 × SEMA3G) + (−0.0912 × GCAT) + (−0.0820 × SFRP2) + (−0.0625 × WNT7B) + (0.0747 × EN1) + (0.0791 × HK2) + (0.0986 × TPX2) + (0.1003 × SMS) + (0.1020 × TFPI) + (0.1184 × CRNDE) + (0.1218 × PLOD3) + (0.1281 × KLF10) + (0.1572 × ACSL1)(2)

The results presented that the distribution of risk-score was greatly distinct in the ICD-related gene clusters. As compared with the ICD-cluster A, cluster B exhibited a lower risk score (Figure 5B), which was similar to the results of the TME scores of ICD-related genes. The results revealed that the low-risk score might indicate low TME scores. The levels of the risk score in the DEG clusters are presented in Figure 5C. According to the median risk score, the patients were grouped into low-risk (*n* = 458) and high-risk (*n* = 458) groups. As shown in Figure 5D,E, the results indicated that the higher risk scores indicated, the higher the death rate and lower the survival time for glioma patients. The PCA analyses showed clear differences in the distribution of low- and high-risk groups (Figure 5F). Furthermore, the K–M curve analysis demonstrated that the low-risk group exhibited a longer survival time as compared to the high-risk group (*p* < 0.001; Figure 5G). Moreover, the ROC curves of 1-, 3-, 5-, and 10-year survival rates were plotted, showing the area under the ROC curve (AUC) values of 0.822, 0.903, 0.896, and 0.870, respectively (Figure 5H). Additionally, the K–M curve analysis of the low- and high-risk groups was combined with the status of radiotherapy in the training set to assess the combined effect of the risk score and radiotherapy (Figure 5I). Notably, between the low- and high-risk groups, the patient who were treated with radiotherapy, showed a better prognosis as compared to those who were not treated with radiotherapy. These results indicated radiotherapy might be more beneficial for the patients of the low-risk group. Consistently, the patients in the low-risk group showed longer survival times as compared with those in the high-risk group. In conclusion, the predictive potential of our risk score was independent of radiotherapy, and the patients with lower risk may benefit more from radiotherapy.

The predictive potential of our risk score was further investigated in the testing set and external CGGA (mRNAseq_325, mRNAseq_693, microarray) sets (Appendix A). All of the patients in these datasets were also divided into two risk groups based on the results from the training set. We presented the results of dot plots and PCA analyses in Appendix A, respectively. As shown in Appendix A, in all of the K–M curve analyses, the low-risk groups presented a better prognosis as compared with the high-risk ones (*p* < 0.001). Immediately, ROC curve analyses of the 1-, 3-, 5-, and 10-year were conducted to verify the predictive potential of the risk score. These results illustrated that our risk score was correlated with a favorable AUC value (Appendix A).

### 3.6. Verification of the Expression Levels of 14-Signature Genes

To assess the expression levels of the 14-signature genes, six tumor samples from patients with glioma, and six matching control samples from patients with intracerebral hemorrhage, and qRT-PCR was performed. In the glioma samples, compared with the control samples, the expression levels of *ACSL1*, *GCAT*, *SEMA3G*, and *WNT7B* were increased, while those of *BMP2*, *EN1*, *GRNDE*, *KLF10*, *PLOD3*, *SFRP2*, *TFPI*, and *TPX2* decreased. Moreover, the expression levels of *HK2* and *SMS* remained unchanged (Appendix A).

### 3.7. Clinical Relevance and Prognostic Risk-Score Stratification Analyses

The correlations of risk-score with the different clinical characteristics, including age, sex, WHO grade, histology, and *IDH* mutation status, were investigated in this study to reveal the effects of risk-score on the clinicopathological features. As shown in Appendix A, this study demonstrated that the patients’ grades changed with the increase in the risk score (*p* < 0.001). In order to further demonstrate if the risk score could be an independent factor for the OS rate of glioma patients, univariate and multivariate Cox regression analyses of the clinicopathological features were conducted in combination with the risk score among the training, testing, CGGA mRNAseq_325, mRNAseq_693, and microarray datasets, respectively (Appendix A). These results demonstrated that both the risk score (*p* < 0.001) and WHO grade (*p* < 0.001) were independent prognostic factors. Furthermore, K–M curve analyses was performed to analyze the different subtypes of clinical features, including gender (male and female), age (≤45 and >45 years), WHO grade (grade II–III and grade IV), and *IDH* mutation (mutation and wild), for assessing their potential to forecast the risk score for the survival time of glioma patients. These results demonstrated that the low-risk patients in all the subtypes of clinical features had a longer survival time as compared to the high-risk patients (*p* < 0.001 in all subsets) (Appendix A). Thus, these results demonstrated our risk score can be applied to most subtypes of clinicopathological features.

### 3.8. Analyzing the Immune Checkpoint and TME between the Risk Subgroups

The number of infiltrated immune cells in each glioma sample was calculated using CIBERSORT analysis. Then, the relationships between the risk score and immune cells were presented using scatter plots, which revealed that the M0, M1, and M2 macrophages, neutrophils, activated memory CD4 + T cells, CD8 + T cells, and follicular helper T cells were positively related to the risk-score, while the activated mast cells, monocytes, resting memory CD4 + T cells, resting dendritic cells, and activated NK cells were negatively related to the risk-score (Figure 6A). In our two risk subgroups, we demonstrated that the major immune cells were greatly different (Appendix A). The TME scores between the high- and low-risk groups revealed that the high-risk group had higher immune, stromal, and ESTIMATE scores (Figure 6B). As shown in Figure 6C, the results suggested that most of the 14 signature-related genes were related to activated mast cells, M0 macrophages, neutrophils, monocytes, and T cells. Moreover, the majority of the 30 immune-checkpoint genes, such as *PDCD1*, *CD274*, and *CTLA-4*, showed significantly lower expression levels in the low-risk group in comparison with the high-risk group (Figure 6D).

### 3.9. Correlation between CSC Index and Risk-Score

In order to reveal the linear correlation between the CSC index and our risk score, a scatter diagram of the glioma patients was plotted. The results demonstrated a strong negative correlation (*R* = −0.54, *p* < 0.001), suggesting that the decrease in risk might also decrease the cellular differentiation levels, while increasing the stem cell characteristics (Figure 7A).

### 3.10. Drug Sensitivity and Mutation Analysis

According to previous studies, a high TMB indicates more neo-antigen sites, thereby showing the improved efficacy of immunotherapy [25]. This study showed that our risk score had a significantly positive correlation with the TMB (*R* = 0.56, *p* < 0.001), suggesting that the high-risk group might better respond to immunological therapy (Figure 7B,C). In addition, somatic mutation analyses were performed to determine the top 20 mutated genes in the low- and high-risk groups. The results showed that the top five mutated genes in the low- and high-risk groups were *IDH1*, *TP53*, *ATRX*, *CIC*, and *FUBP1*, and *TP53*, *PTEN*, *EGFR*, *TTN*, and *IDH1*, respectively (Figure 7D,E). Notably, although there were differences in the top five mutated genes, both *IDH1* and *TP53* were highly mutated in the two datasets. Additionally, the sensitivity of the glioma patients to chemotherapy was assessed in the two groups. The IC50 values of bleomycin, cisplatin, etoposide, methotrexate, paclitaxel, and vinblastine were higher, while that of imatinib was lower in the low-risk group in comparison with the glioma patients of the high-risk group (Figure 7F–L). In short, these results demonstrated that the low-risk group might be more sensitive to most glioma chemotherapeutic agents, which might contribute to the better prognosis of the low-risk group.

### 3.11. Constructing a Prognostic Nomogram

In order to enhance and optimize the clinical application of our risk score model, a nomogram was constructed to predict the 1-, 3-, 5-, and 10-year survival rates of patients, characterized by the risk score and WHO grades (Figure 8A). The AUC values of the ROC curves demonstrated that our nomogram model could efficiently predict the 1-, 3-, 5-, and 10-year survival rates of patients in all of the datasets (Figure 8B–F). Furthermore, the ROC curve analyses of the nomogram and WHO grade at 1-, 3-, 5-, and 10-year survival rates were respectively performed to compare their predictive potentials in the five datasets. The result showed that, in the training group, the AUC values of 1-, 3-, 5-, and 10-year survival rates curves of the nomogram were 0.838, 0.895, 0.894, and 0.848, respectively, while those of the WHO grade were 0.838, 0.895, 0.894, and 0.848, respectively (Appendix A). These indicated that all AUC values of the nomograms in training datasets were higher as compared with those of the WHO grade. The other four datasets, including testing (Appendix A), CGGA mRNAseq_325 (Appendix A), mRNAseq_693 (Appendix A), and microarray (Appendix A), showed similar results. These results suggested that our nomogram model had better prognostic potential as compared with the WHO grade. This finding was subsequently verified in all five datasets using the calibration plots (Figure 8G–K).

## 4. Discussion

ICD, as demonstrated in previous studies, can activate various immune cells and is a significant part of the TME and antitumor immune response [1,2,7,26,27]. However, the studies on glioma have mainly focused on a single ICD-related gene or a single cell type in the TME. Therefore, it is critical to study the overall effects of ICD-related genes, TME, and the infiltration of immune cells. The present study investigated ICD-related genes at both the transcription (mRNA) and genetic (CNVs) levels in glioma. First, two entirely different ICD-related gene clusters were identified, and, in comparison with cluster B, the patients in cluster A showed worse clinical characteristics and prognosis. Moreover, the TME scores were also significantly different in both clusters. Meanwhile, the GSVA analysis of these two gene clusters revealed that the cluster A genes were significantly enriched in immune-related pathways, such as NK cell-mediated cytotoxicity, antigen processing and presentation, cytokine receptor interaction, leukocyte trans-endothelial migration, NOD-like, and Toll-like receptor signaling pathways, while the cluster B genes were negatively enriched in these pathways. Moreover, based on the ICD-related gene cluster analysis, a set of DEGs was obtained and GO and KEGG enrichment analyses were performed on these DEGs, which illustrated that the DEGs were also closely correlated with the genesis, activation, and differentiation in immune cells. Based on these DEGs, the DEG clusters were generated. The K–M curves, univariate and multivariate Cox regression analyses and clinical correlational analysis demonstrated that the gene clusters could independently predict the clinicopathological features and prognosis of glioma patients. Therefore, a prognostic signature, containing 14 genes, was filtered out to develop the risk-score model. The expression levels of these signature-related genes were further assessed using qRT-PCR. Then, we further verified the prediction potential of our risk score. Our results demonstrated that the ICD-related gene clusters A and B were correlated with higher and lower risk scores, respectively. Numerous clinicopathological features, such as OS, somatic mutation rates, clinical features, TME, etc., between the two risk-score groups showed significant differences. Finally, we constructed a nomogram model by combining our risk score and WHO grade for a more convenient and efficient clinical application of the risk-score model. In conclusion, the prognostic model can evaluate the prognosis and predict the treatment effects in patients with glioma, and can help in understanding the pathogenesis and progression of glioma.

Glioma is the most malignant brain tumor in adults. Conventional chemoradiotherapy is not effective in prolonging the survival times of glioma patients. Currently, antitumor immunotherapy has been widely used for numerous tumor types, including glioma. However, the prognosis of glioma patients is different after immunotherapy [28]. Therefore, the TME in glioma should be fully understood to increase the efficacy of immunotherapy. The TME is mainly composed of tumor cells, immune cells, surrounding inflammatory cells, tumor-related fibroblasts, nearby stromal tissues, micro-vessels, and different cytokines and chemokines. The majority of immune cells are macrophages, granulocytes, and lymphocytes, which are the significant components in antitumor immune responses [29,30,31]. Numerous studies revealed that TME was connected with the biological progression and treatment resistance of tumors [31,32]. In this study, the ICD-related genes were clustered into clusters A and B. Cluster A was characterized by a higher risk score and was positively correlated with the activation of immune-related pathways, while cluster B was characterized by a lower risk score and was negatively correlated with the activation of immune-related pathways. Moreover, the TME scores and number of infiltrated immune cells were significantly different between these clusters and risk groups, indicating that the ICD-related genes might participate in the biological progression of glioma. Numerous studies demonstrated that neutrophil infiltration was positively correlated with the grade and progression of glioma [33,34]. These and other studies also revealed that neutrophil infiltration could predict the prognosis of glioblastoma patients [33,34,35,36]. Additionally, Treg cells could inhibit the antitumor immune response, and their infiltration levels were negatively related to the prognosis of tumor patients [37,38,39]. Notably, the present study discovered that the cluster A and high-risk groups, characterized by a shorter survival time, exhibited higher infiltration of neutrophils and Treg cells, which were consistent with previous studies and indicated that these two immune-cell types could promote the progression of glioma. Numerous studies demonstrated that B cells could prolong the survival time of tumor patients and participate in the antitumor immune responses [40,41]. Consistent with these previous studies, the longer survival time groups, cluster B and low-risk groups, were infiltrated with higher levels of naive and memory B cells as compared to the cluster A and high-risk groups, which indicated that B cells could inhibit the development of glioma. The macrophages in TME are divided into M1 and M2 macrophages [29,31,42]. M1 macrophages can induce the Th1 immune response and phagocytosis, thereby killing the tumor cells and inhibiting their development, while M2 macrophages release cytokines and mediate the Th2 immune response, thereby promoting immune suppression and tumor evolution [39,43]. In this study, both M1 and M2 macrophages were greatly elevated in the cluster A and high-risk groups, but the number of M2 macrophages was significantly higher as compared with that of the M1 macrophages. This suggested that the M2 macrophage was the major macrophage subset in cluster A and the high-risk group, indicating the complexity of TME in glioma.

Due to the limitations in conventional chemoradiotherapy, tumor immunotherapy has been widely studied and applied in clinical practices as a new tumor treatment strategy. Immune checkpoint inhibitors (ICIs) are a significant component of tumor immunotherapy. Numerous studies have widely applied ICIs, targeting the *CTLA-4*, *PDCD1*, and *CD274*, in preclinical and clinical studies to explore their effectiveness and safety [44]. Similarly, ICIs have also shown great success in treating glioblastoma [44,45]. The high expression levels of *PDCD1* show the better clinical efficacy of *PDCD1*/*CD274* checkpoint inhibitors [46,47,48]. Numerous studies have shown significantly higher levels of *PDCD1* in glioblastoma tissues as compared with normal brain tissues [49,50]. Moreover, the combination therapy of targeting *CTLA-4* and *PDCD1* could better suppress tumor progression [45,51]. Notably, the expression levels of *PDCD1*, *CD274*, and *CTLA-4* significantly increased in the cluster A and high-risk groups, indicating the better efficacy of *PDCD1*, *CD274*, and *CTLA-4* inhibitors in these two groups.

There were certain limitations to this study. First, all the transcriptional, genetic, and clinicopathological data were acquired from common databases and lacked data from the patients in our department. Therefore, any selection bias in these downloaded data might have caused significant outcome deviations. Second, this was a retrospective study; therefore, more prospective and biological studies are needed to verify these findings. Moreover, several related clinicopathological features, such as neo-adjuvant chemoradiotherapy and the details of surgery and drugs, were not included in the data, which might also affect the results of ICD and immunotherapy.

## 5. Conclusions

Based on the 31 ICD-related genes, the genes in the datasets were divided into two ICD-related gene clusters and two ICD-related DEG clusters. Then, a risk-score system was developed to assess the OS rate and TME scores of the patients with glioma. Furthermore, the potential regulatory mechanisms in these clusters and the correlations between TME and ICD in glioma were investigated. Moreover, the correlations among ICD-related genes, immunotherapy, and ICIs were also assessed. In conclusion, our ICD-related risk score may become an excellent predictive tool for the survival time and therapeutic efficacy of patients with glioma, and may be conducive to individualized immunotherapy in glioma patients.

## Figures and Tables

**Figure 1 cancers-14-05665-f001:**
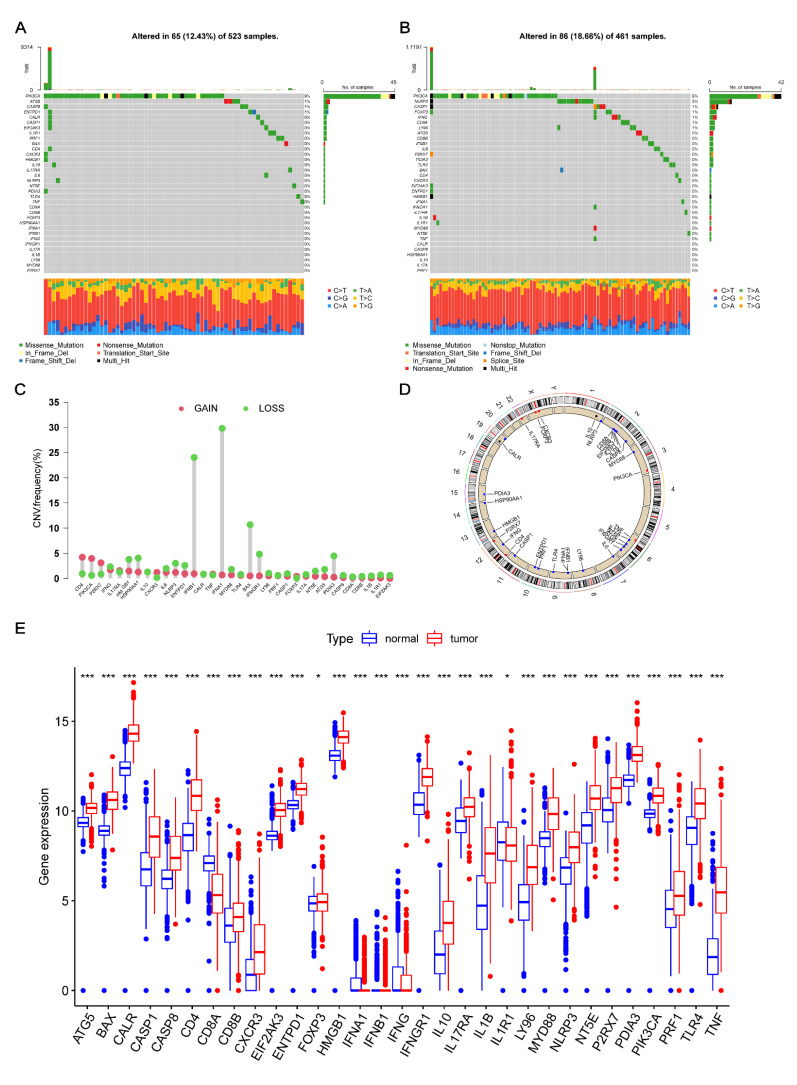
Transcriptional changes in the ICD-related genes of glioma. (**A**,**B**) Mutation frequencies of 31 ICD-related genes in 523 and 461 patients with LGG and GBM, respectively, from the TCGA database. (**C**) Increase, decrease, and no changes in CNVs among the ICD-related genes. (**D**) Position of ICD-related genes on the 23 pairs of chromosomes. (**E**) Analysis of the expression levels for 31 ICD-related genes between normal and glioma tissues. *, *p* = 0.01–0.05; ***, *p* < 0.001; ICD, immunogenic cell death; LGG, lower-grade glioma; GBM, glioblastoma; TCGA, the Cancer Genome Atlas; CNV, copy number variety.

**Figure 2 cancers-14-05665-f002:**
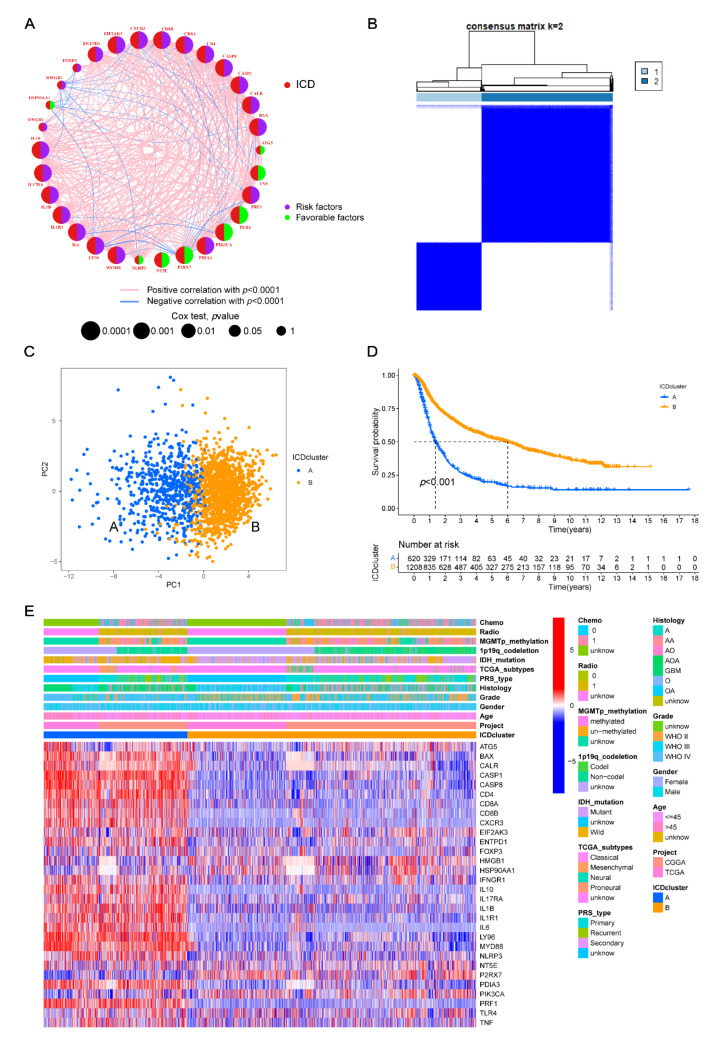
ICD-related gene clusters and clinicopathological and biological features of two clusters. (**A**) Interactions among ICD-related genes in glioma. The curves connecting the ICD-related genes indicate their relationships; the width of the curves indicates relationship strength. The blue and red colors indicate negative and positive relationships, respectively. The circle size represents the P-value. (**B**) Matrix heatmap, revealing two clusters (k = 2). (**C**) PCA analyses, indicating a significant difference between the two clusters. (**D**) Kaplan–Meier curves, revealing a notable difference in OS between clusters A and B. (**E**) Heatmap of clinical characteristics and two clusters based on expression levels of 31 ICD-related genes. ICD, immunogenic cell death; PCA, principal components analysis; OS, overall survival.

**Figure 3 cancers-14-05665-f003:**
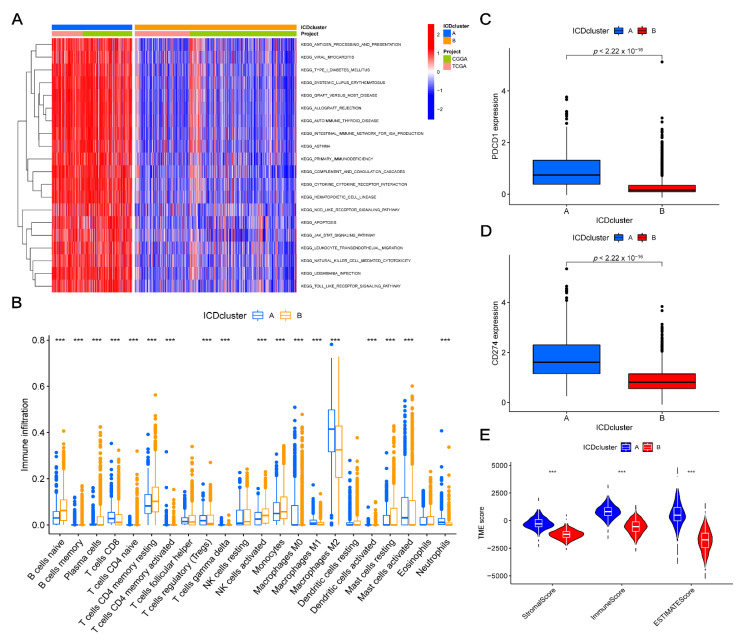
Correlations between the TME and two ICD-related gene clusters. (**A**) GSVA between the two clusters for biological pathways. The red indicates positive correlation, and the blue indicates negative correlation. (**B**) Differences in the 22 immune cells between clusters A and B. (**C**,**D**) Difference in the PDCD1 and CD274 expression levels between clusters A and B, respectively. (**E**) Relationships between two clusters and TME scores. ***, *p* < 0.001; ICD, immunogenic cell death; GSVA, gene set variation analysis; TME, tumor microenvironment.

**Figure 4 cancers-14-05665-f004:**
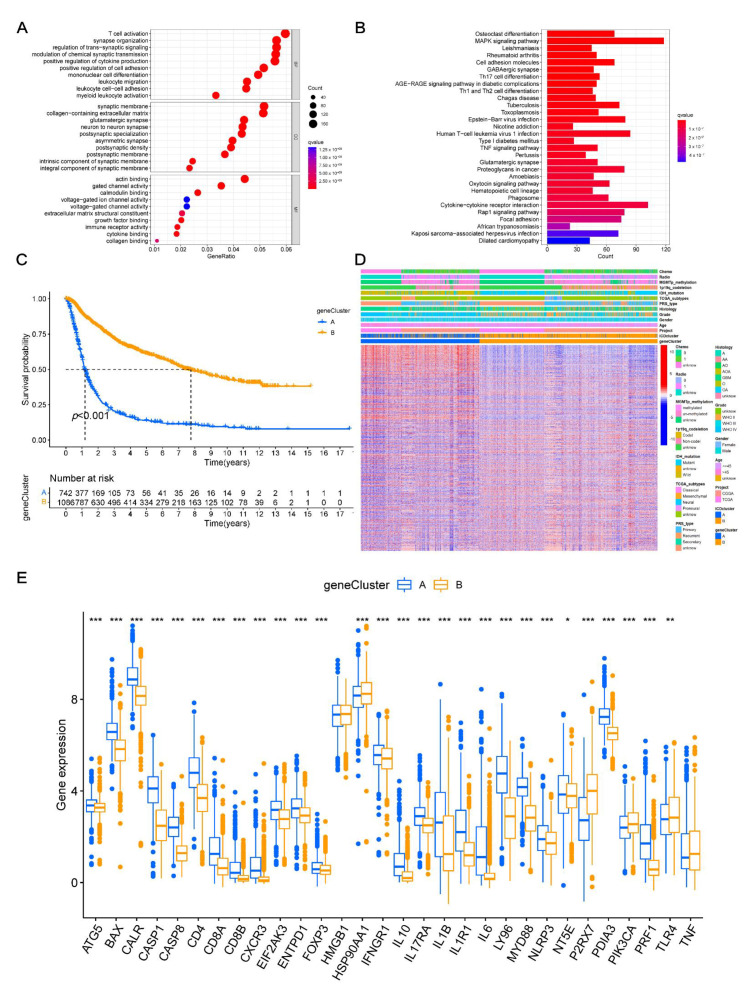
Development of DEGs clusters based on DEGs. (**A**,**B**) GO and KEGG analysis of ICD-related gene clusters. (**C**) Kaplan–Meier curves for two DEGs clusters to assess the OS (*p* < 0.001). (**D**) Relationships between clinical characteristics and two DEGs clusters. (**E**) Differences in the 31 ICD-related genes expression levels between the two DEGs clusters. *, *p* = 0.01–0.05; **, *p* = 0.001–0.01; ***, *p* < 0.001; ICD, immunogenic cell death; DEGs, differentially expressed genes; GO, Gene Ontology; KEGG, Kyoto Encyclopedia of Genes and Genomes; OS, overall survival.

**Figure 5 cancers-14-05665-f005:**
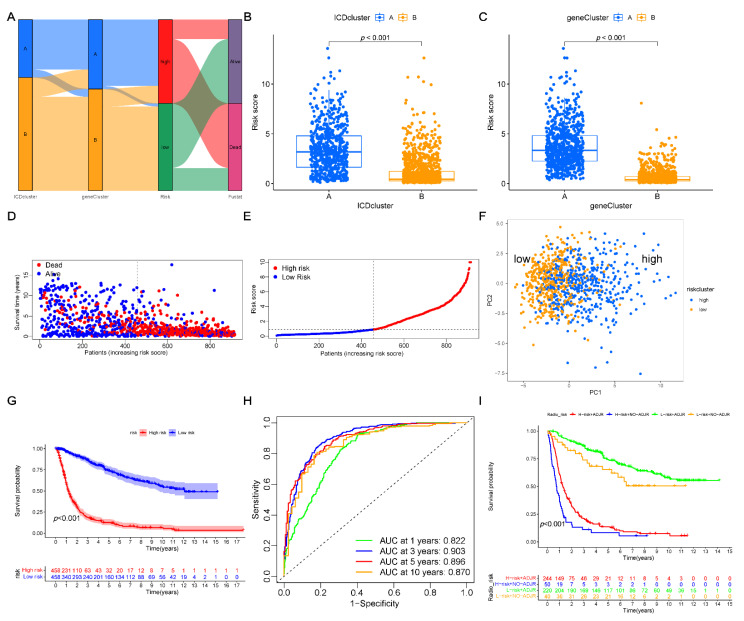
Development of the risk score system using the training set. (**A**) Sankey diagram presenting the distribution of different subtypes. (**B**) Difference in the risk scores of two ICD-related gene clusters. (**C**) Difference in the risk scores of two DEGs clusters. (**D**,**E**) Ranked dots and scatter plots, presenting the distribution of risk and survival outcomes of patients. (**F**) PCA analyses, indicating a significant difference between the two risk groups. (**G**) Kaplan–Meier curves for the two risk groups to assess the OS (*p* < 0.001). (**H**) ROC curves of the risk score to assess the sensitivity and specificity of 1-, 3-, 5-, and 10-year survival rates. (**I**) Kaplan–Meier curves for four groups combining risk score and adjuvant radiotherapy to assess the OS (*p* < 0.001). ICD, immunogenic cell death; DEGs, differentially expressed genes; ADJC, adjuvant radiotherapy; PCA, principal component analysis; OS, overall survival; ROC, receiver operating characteristic.

**Figure 6 cancers-14-05665-f006:**
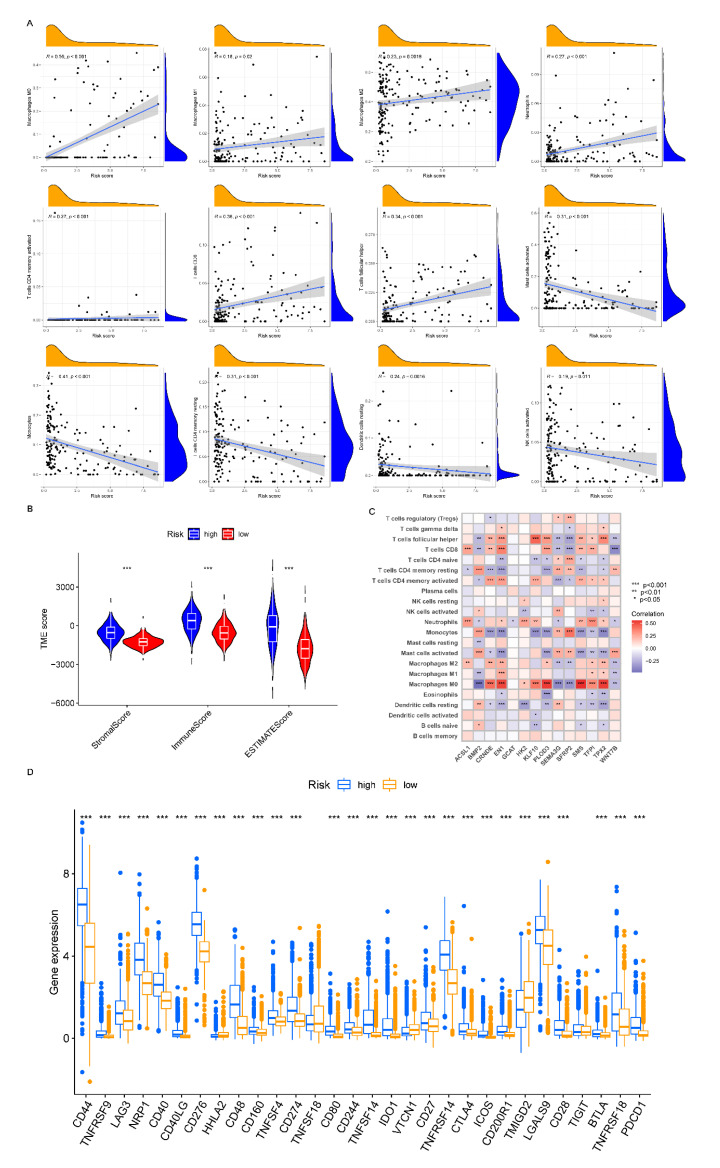
Analysis of TME and checkpoint genes between the different risk groups. (**A**) Relationships between risk scores and 12 immune cells. (**B**) Relationships between risk groups and TME scores. (**C**) Heatmap of the relationships among 22 immune cells and 14 signature genes. (**D**) Differences in the immune checkpoint genes of the two risk groups. *, *p* = 0.01–0.05; **, *p* = 0.001–0.01; ***, *p* < 0.001; TME, tumor microenvironment.

**Figure 7 cancers-14-05665-f007:**
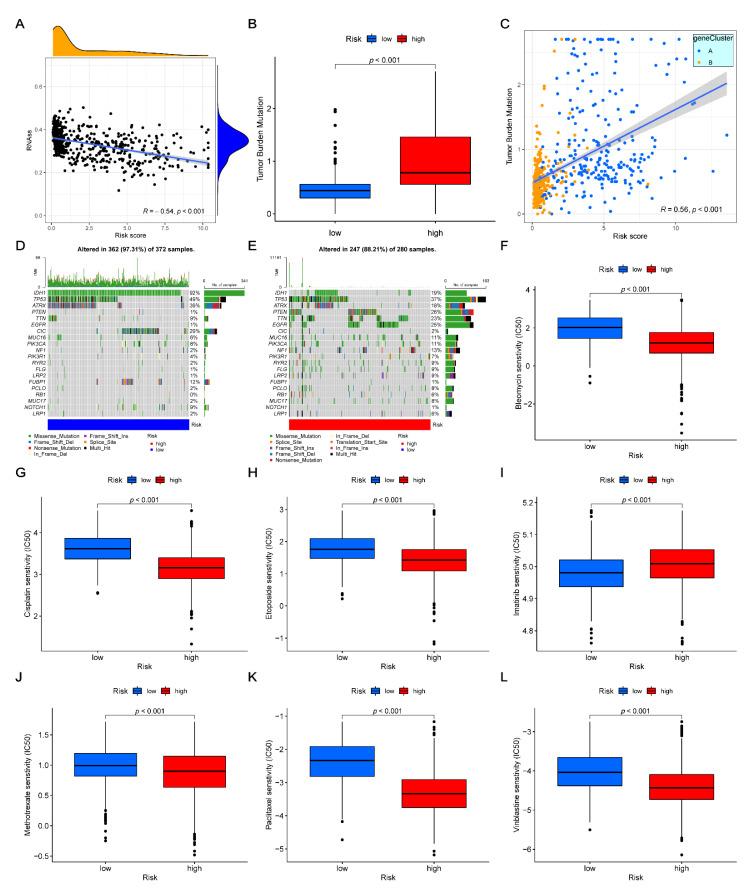
Overall evaluations of our risk scores in glioma. (**A**) Correlations between risk scores and CSC index. (**B**) Difference in the TMB of high- and low-risk groups. (**C**) Correlations of risk scores and TMB. (**D**,**E**) Somatic mutation analyses in two risk groups. (**F**–**L**) Correlations between the different risk groups and sensitivity of chemotherapeutic agents. CSC, cancer stem cell; TMB, tumor mutation burden.

**Figure 8 cancers-14-05665-f008:**
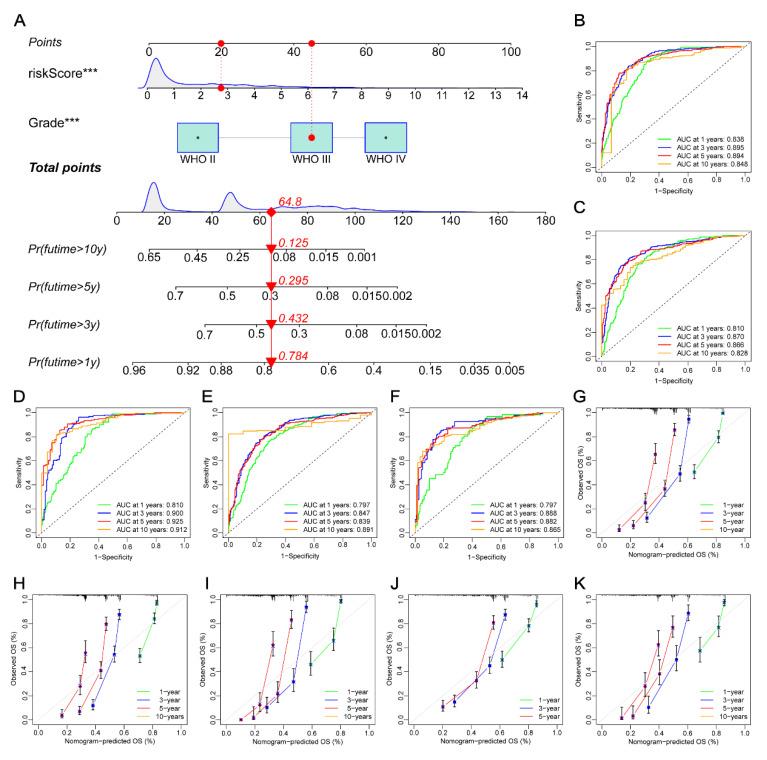
Development and verification of our nomogram. (**A**) Nomogram in glioma patients for assessing the 1-, 3-, 5-, and 10-year OS in the training dataset. (**B**–**F**) ROC curves of 1-, 3-, 5-, and 10-year in the training, testing, CGGA mRNAseq_325, mRnaseq_693, and microarray datasets. (**G**–**K**) Calibration curves of 1-, 3-, 5-, and 10-year in the training, testing, CGGA mRNAseq_325, mRnaseq_693, and microarray datasets. ***, *p* < 0.001; OS, overall survival; ROC, receiver operating characteristic; CGGA, the China Glioma Atlas.

**Table 1 cancers-14-05665-t001:** Multivariate Cox analyses of 12 ICD-related genes correlated to prognosis in glioma patients.

Id	HR	95%CI	*p*-Value
*BAX*	1.1447	1.0528~1.2447	0.0016
*CALR*	1.1871	1.0917~1.2909	*p* < 0.001
*CASP1*	1.1335	1.0664~1.2049	*p* < 0.001
*CASP8*	1.2092	1.1051~1.3232	*p* < 0.001
*CD4*	1.1065	1.0368~1.1808	0.0023
*IL10*	1.1363	1.0367~1.2455	0.0063
*IL1R1*	1.0916	1.0307~1.1560	0.0028
*IL6*	1.0610	1.0134~1.1109	0.0115
*LY96*	1.0940	1.0451~1.1453	*p* < 0.001
*MYD88*	1.2355	1.1433~1.3351	*p* < 0.001
*NT5E*	0.8990	0.8479~0.9531	*p* < 0.001
*PDIA3*	1.2042	1.0773~1.3459	0.0011

## Data Availability

The transcriptome profiling (transcripts per kilobase million, TPM) and related clinical data of glioma were downloaded from the China Glioma Atlas (CGGA, http://www.cgga.org.cn/, (accessed on 5 May 2022)) and the Cancer Genome Atlas (TCGA, https://portal.gdc.cancer.gov/, (accessed on 4 May 2022)) databases. And the data related with our study have been uploaded in Appendix A.

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
