# Peer review of "Identification of Immunogenic Cell Death-Related Signature for Glioma to Predict Survival and Response to Immunotherapy"

_cancers, 2022, doi:10.3390/cancers14225665_

Round 1

Reviewer 1 Report

The genetic, transcriptional, and clinical data of glioma samples were acquired from the five distinct databases and analyzed in terms of genes and transcription levels.  The correlations between risk groups and prognosis, cells in the tumor microenvironment (TME) and immune cells infiltration, chemosensitivity and cancer stem cell (CSC) index were assessed. This study analyzed the ICD-related genes in glioma and evaluated their role in the overall survival, clinicopathological characteristics, TME and immune cell infiltration of glioma.  A highly precise nomogram model was constructed to enhance and optimize the clinical application of the risk score. The results demonstrated that the risk score could independently predict the overall survival rate and immunotherapeutic response of glioma patients. It is suggested that the ICD-related risk score might become a predictive tool for the survival time and therapeutic efficacy of patients with glioma with the goal of developing better immunotherapeutic strategies, although this is speculative.  This paper represents a tremendous amount of work.

Author Response

Dear reviewer,

    Thank you so much for your reviewing! We deeply appreciate your recognition of our research work.

    Special thanks to you for your good comments.

Yours sincerely,

Zhiqiang Sun

Reviewer 2 Report

The authors have done a commendable job of comprehensive characterization of ICD-related genes in glioma. The methodology and analysis are detailed and the results are interesting.

Author Response

(The authors gave the same response as above.)

Reviewer 3 Report

Zhiqiang Sun and coworkers describe the importance to identify a ICD gene signature useful in selecting the best terapeutic strategy for glioma. Improvement in the management of patients with glioma is still needed as there is not yet a truly effective therapeutic strategy. Thus, the manuscript is interesting. However, two papers have recently been published in Frontiers Immunology on ICD risk signature in glioma (Front Immunol. 2022 Oct 17;13:1011757. doi: 10.3389/fimmu.2022;  Front Immunol. 2022 Sep 26;13:992855. doi: 10.3389/fimmu.2022.992855)

Authors need to highlight and clarify how their findings are new and original to the scientific community respect to the previously published data. Without this kind of dicussion, the manuscript lacks of originality.

Author Response

Dear Reviewer,

    Thank you for your comments concerning our manuscript entitled “Identification of immunogenic cell death-related signature for glioma to predict survival and response to immunotherapy” (ID: cancers-2022219). Those comments are all valuable and very helpful for revising and improving our paper, as well as the important guiding significance to our researches. After receiving your comments, we have carefully read the two articles you mentioned about ICD and studied comments carefully. Revised portion are used the “Track Changes” function and marked in red in the paper. The main corrections in the paper and the responds to your comments are as following:

  1. We note that “Front Immunol. 2022 Oct 17; 13:1011757. doi: 10.3389/fimmu.2022” is to study the prognostic predictive ability of ICD risk signature in low-grade gliomas; “Front Immunol. 2022 Sep 26; If 92855. Doi: 10.3389 / fimmu. 2022.992855” is the study of the prognostic predictive ability of ICD risk signature in GBM. The prognostic value of ICD risk signature in whole glioma has not been investigated. We note that glioma is the most recurrent tumor in the central nervous system, and its recurrence is often accompanied by an increase in tumor grade. Therefore, we believe that it is insufficient to study LGG or GBM alone to predict the prognosis of patients with glioma. We systematically investigated the prediction ability of ICD risk signature for prognostic and immune response in grade II, III and IV gliomas by integrating all LGG and GBM data from TCGA database. Therefore, compared with study of a single subtype in glioma, our risk model can predict the effect more accurately.

  1. Compared to the two studies mentioned above, our approach to constructing the risk model was different. In the two studies mentioned above, risk models were constructed by selecting several genes associated with the prognosis of patients from ICD-related genes. However, the accuracy of models constructed in this way is limited. Our approach to risk modeling was not limited to the 31 ICD-related genes, but to first verify that the prognosis of patients with two ICD-related clusters is significantly different, and then we found 3517 differentially expressed genes between the two clusters. Of the 3517 differentially expressed genes, Multivariate survival analysis and lasso regression were used to find out the hub genes that could best predict the prognosis of patients, and then a risk model was constructed. Using our approach, we were able to identify ICD-related hub genes that best predicted patient outcomes and significantly improve the predictive power of risk models.

  1. We compared our risk model with WHO grade in predicting the overall survival (OS) rate of glioma patients. The results showed that our risk model was superior to WHO grade in predicting the OS rate of glioma patients. This was lacking in the two studies.

  1. We noted that the predictive power of our risk model for 1 -, 3 -, 5 -, and 10-year survival rates of glioma patients was significantly higher than that of the above two studies. This further proves the accuracy of our constructed model.

  1. We found that in the above two studies, they only investigated the predictive power of the risk model for the response to immunotherapy, while in our study, we not only evaluated the predictive power of the model for immunotherapy, but also analyzed its predictive power for the response to radiotherapy in glioma patients.

    In response to your suggestion, we have reiterated the importance, innovation and originality of our work in the section of introduction and marked them.

    Special thanks to you for your good comments.

    If there is anything else, we should do, please do not hesitate to let us know. Thank you and best regards.

Yours sincerely,

Zhiqiang Sun

Round 2

Reviewer 3 Report

Given that the authors add a sentence in the introduction to better highligh the novelty of their manuscript, they should then include the two recent papers published in Front Immunol 2022 in the reference list.

Author Response

Dear Reviewer,

    Thank you for your comments concerning our manuscript entitled “Identification of immunogenic cell death-related signature for glioma to predict survival and response to immunotherapy” (ID: cancers-2022219). Those comments are all valuable and very helpful for revising and improving our paper, as well as the important guiding significance to our researches. After receiving your comments, we have made corresponding changes in the manuscript, cited the two recent papers published in Front Immunol 2022 and included them in the reference list (line 59 of page 2 and line 788~794 of page 24). Revised portion are used the “Track Changes” function and marked in red in the paper.

    Special thanks to you for your good comments.

    If there is anything else, we should do, please do not hesitate to let us know. Thank you and best regards.

Yours sincerely,

Zhiqiang Sun
